# Enhancing Precision Drug Recommendations via Fine-grained Exploration of Motif Relationships

## Abstract

Making accurate and safe drug recommendation for patients has always been a challenging task. Even though *rule-based, instance-based*, and *longitudinal data-based* approaches have made notable strides in drug modeling, they often neglect to fully leverage the rich motifs information. However, it is widely acknowledged that motifs exert a significant influence on both drug action and patient symptomatology. Therefore, there is a pressing need for more comprehensive exploration this invaluable information to further enhance drug recommendation systems. To tackle the aforementioned challenges, we present *DEPOT*, a novel drug recommendation framework that leverages motifs as higher-level structures to enhance recommendations. In our approach, we employ chemical decomposition to partition drug molecules into motif-trees and design a structure-aware transformer for drug representation, enabling us to capture the structural information among substructures. To investigate the relationship between disease progression and motifs, we conduct a meticulous exploration from two perspectives: repetition and exploration. This comprehensive analysis allows us to gain valuable insights into the drug turnover, with the former focusing on reusability and the latter on discovering new requirements. Furthermore, we incorporate historical DDI effects and employ a nonlinear optimization objective to stabilize the training process, ensuring the safety of recommended drug combinations. Extensive experiments conducted on two data sets validate uniqueness and efficacy of the *DEPOT*.

## 1 Introduction

With the advancement of deep learning technology and the availability of vast medical data, artificial intelligence (AI) has emerged as a powerful ally in medical decision-making (Schimunek et al., 2022). Among the diverse range of applications, drug recommendation emerges as a field of paramount importance. Its main goal revolves around comprehending the inherent interplay among diseases, diagnostic procedures, and medications, ultimately facilitating the personalized prescription of treatments tailored to individual patient conditions—an aspect of healthcare that holds exceptional significance (Sun et al., 2022a). Notable examples in this field include Baidu's Medical Brain, Berkeley Lights, and Google Health, all utilizing AI-driven drug recommendation systems, as exemplified in (Shaheen, 2021).

Unlike conventional recommendation algorithms (Wu et al., 2022a), drug recommendation encounters distinct challenges associated with safety requirements and personalized suggestions (Yang et al., 2021b). The early rule-based approach initially depended on experienced physicians manually formulating rules to guide the diagnostic process (Chen et al., 2016). Nonetheless, this method proved to be time-consuming, labor-intensive, and reliant on extensive expert knowledge and a substantial rule database for effective decision-making. The instance-based method draws inspiration from collaborative filtering algorithms, seeking to uncover co-occurrence relationships between drugs and symptoms (Zhang et al., 2023). However, its efficacy in discerning individual patient needs may be limited, primarily relying on commonalities observed in the broader population. More recently, longitudinal approaches have emerged, incorporating patient history to a greater extent (Yang et al., 2023). This longitudinal observation and analysis offer a comprehensive grasp of the patient's condition, facilitating the provision of precise diagnosis, as depicted in Figure 1(a).

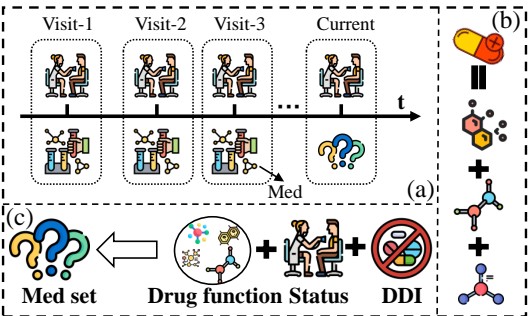

Figure 1: (a) is the patient's longitudinal visit and prescription process (b) illustrates drug is composed of multiple motifs (c) implies drug recommendations need to take into account patient status, drug function, and DDI effects.

While significant progress has been made in these methods, notable challenges persist. First, the substructures of drugs, referred to as motifs, often remain underexplored despite embodying the essence of drug functionality, as exemplified in Figure 1(b). For example, in the case of nonsteroidal anti-inflammatory drugs (NSAIDs), the *phenyl ring* and *acetic acid group* (two common motifs) synergistically form a specific pharmacophore. This pharmacophore engages with the target enzyme cyclooxygenase, leading to the inhibition of prostaglandin synthesis, thereby conferring anti-inflammatory, analgesic, and antipyretic effects (Lu et al., 2020). However, traditional methods typically represent drugs as SMILES (Simplified Molecular Input Line Entry System) sequences (Jiang et al., 2022) or atomic connectivity graphs (Wu et al., 2023b), often overlooking the comprehensive exploration and modeling of these intricate synergistic relationships among motifs. Although some studies have touched upon motifs (Yang et al., 2023), they fall short of capturing the intricate structural interactions between them, thereby impeding the accuracy of drug representation and recommendation. Secondly, current approaches lack explicit modeling of the dynamic relationship between drug motifs and the evolution of a patient's medical condition, even though capturing this relationship is crucial, as evidenced in cancer treatment (Peerzada et al., 2021; Yang et al., 2021c). For instance, specific motifs in anti-cancer drugs interact with signaling pathways within tumor cells, effectively inhibiting their growth, necessitating the continuous administration of drugs over a specified period. However, disease progression can lead to mutations and drug resistance (Crucitta et al., 2022; Yang et al., 2021a), compromising the interaction between drug motifs and tumor cells and calling for the exploration of new drug motifs as replacements. Therefore, it is important to redirect efforts toward better integrating motif collaboration and accurately capturing the dynamic interplay between drug motifs and disease progression, as depicted in Figure 1(c).

To address the aforementioned challenges, we present a novel method to *enhance precision Drug rEcommendation via fine-grained exPloration of mOtif relaTionships*, namely ***DEPOT***. Our approach harnesses specific chemical knowledge (Jiang et al., 2023) to transform each drug molecule into a **motif-tree**, and associates it with the proposed **structure-aware graph transformer** that effectively captures long-distance collaboration effects between motifs. This motif-level perspective alleviates the low semantic richness of previous atomic-level aggregation as well as facilitates a more comprehensive understanding of structural dependencies within the molecule. Secondly, to model the dynamic relationship between patient disease conditions and drug motifs, we adopt two key perspectives. On the one hand, we design a **repeat encoder** that measures the extent to which past drugs remain functionally relevant by computing Gaussian distributions of disease progression. This enables us to assess the degree to which disease progression necessitate the continuation of specific drug functionalities. On the other hand, we introduce an **explore encoder** that computes probability using all available motifs, facilitating the discovery of novel drug options. Thirdly, we delve deep into the Drug-Drug Interaction (DDI) effect (Dong et al., 2023) and leverage **historical DDI records** to reconstruct the DDI loss function in a nonlinear fashion. This innovative approach not only stabilizes the training process but also strengthens safety considerations in drug recommendation by accounting for the historical DDI information.

In summary, our work has the following contributions: (1) To the best of our knowledge, *DEPOT* stands as the pioneering drug recommendation framework that is based on motifs relations. Unlike previous approaches, we not only consider the structural information and collaboration among drug motifs, but also delve into the dynamic relationship between motifs and disease progression from two distinct perspectives: repetition and exploration. (2) We offer a fresh perspective on tackling

the drug-drug interaction (DDI) problem. We introduce historical DDI effect sequences for the first time, presenting an innovative solution to address safety concerns in drug recommendation with stability and efficiency. (3) We conduct rigorous experimental validation on two extensive public data sets, showcasing substantial advancements achieved by our algorithm in terms of safety and accuracy. We have released the code and data sets [1] for follow-up and reproducibility.

## 2 PROBLEM FORMULATION

**EHR data:** Suppose we have electronic health record (EHR) data, where each patient visit record is denoted as $\mathcal{X}^{(k)} = (\mathbf{x}_1^{(k)}, \mathbf{x}_2^{(k)}, ..., \mathbf{x}_{\mathcal{N}_k}^{(k)})$. Here, $k \in \mathcal{N}$ represents the patient index, and $\mathcal{N}_k$ represents the total number of visits for that patient. Each visit $\mathbf{x}_i^{(k)}$ can be represented as a triplet: $\mathbf{x}_i^{(k)} = (\mathbf{d}_i^{(k)}, \mathbf{p}_i^{(k)}, \mathbf{m}_i^{(k)})$, where $\mathbf{d}, \mathbf{p}$, and $\mathbf{m}$ denote multi-hot diagnosis, procedure, and medication records, respectively. The variables $\mathbf{d}_i^{(k)} \in \{0,1\}^{|\mathcal{D}|}$, $\mathbf{p}_i^{(k)} \in \{0,1\}^{|\mathcal{P}|}$, and $\mathbf{m}_i^{(k)} \in \{0,1\}^{|\mathcal{M}|}$, where $\mathcal{D}, \mathcal{P}, \mathcal{M}$ represent the set of each object, and $|\cdot|$ represents the size of the set.

**Safety Criterion:** The drug-drug interaction (DDI) graph, denoted as $G = \{\mathcal{V}, \mathcal{E}\}$, represents the strong side effects between different drugs. Here, $\mathcal{V} = \mathcal{M} = \{m_1, ..., m_{|\mathcal{M}|}\}$ denotes the set of all drugs, and $\mathcal{E}$ represents the known side effects between pairs of drugs. To elucidate the connections within $\mathcal{E}$, we utilize an adjacency matrix $\mathcal{A} \in \{0,1\}^{|\mathcal{M}| \times |\mathcal{M}|}$, where $\mathcal{A}_{i,j} = 1$ if there exists a documented side effect between drug $i$ and drug $j$. Recommendations that involve a higher number of drug side effects are deemed less reliable. Therefore, our objective is to minimize the drug-drug interaction (DDI) rate, which indicates a lower occurrence of side effects, in order to enhance the quality of recommendations. Now, we can provide a formal definition of *DEPOT*.

**Medical Recommendation:** *Given the patient's historical electronic medical records* $\mathbf{d}_{his}^{(k)} = (\mathbf{d}_1^{(k)}, ..., \mathbf{d}_{\mathcal{N}_k-1}^{(k)})$, $\mathbf{p}_{his}^{(k)} = (\mathbf{p}_1^{(k)}, ..., \mathbf{p}_{\mathcal{N}_k-1}^{(k)})$, *and* $\mathbf{m}_{his}^{(k)} = (\mathbf{m}_1^{(k)}, ..., \mathbf{m}_{\mathcal{N}_k-1}^{(k)})$, *along with the current state* $(\mathbf{d}_{\mathcal{N}_k}^{(k)}, \mathbf{p}_{\mathcal{N}_k}^{(k)})$ *and the DDI graph G, our objective is to recommend a drug combination* $\hat{\mathbf{m}}_{\mathcal{N}_k}^{(k)}$ *while maintaining a low drug-drug interaction (DDI) rate. Here,* $\hat{\mathbf{m}}_{\mathcal{N}_k}^{(k)} \in \{0,1\}^{|\mathcal{M}|}$ *represents the recommended drug combination, where* $|\mathcal{M}|$ *denotes the size of the drug collection.*

Important mathematical notes can be found in Appendix A.

## 3 PROPOSED METHOD

In this section, we outline the proposed framework *DEPOT*, whose architecture is shown in Figure 2.

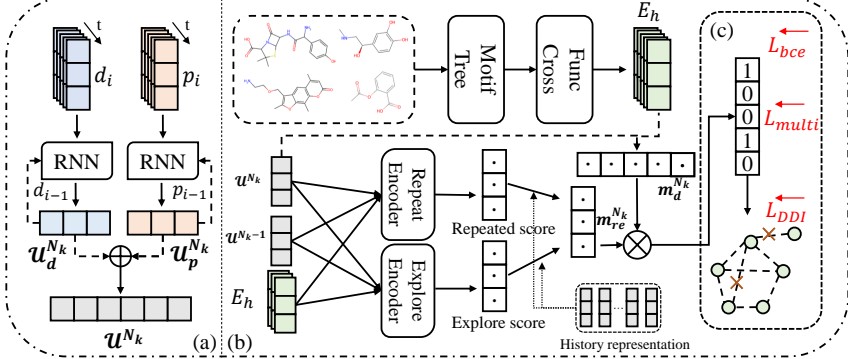

Figure 2: Overall framework of *DEPOT*. It contains (a) a longitudinal patient representation module, (b) a motif-aware drug representation module, and (c) a drug recommendation module.

### 3.1 LONGITUDINAL PATIENT REPRESENTATION

Longitudinal temporal data contains richer semantic information compared to using only current visit records, facilitating a better understanding of personalized disease progression.

---

[1]https://anonymous.4open.science/r/DrugRec

**Embedding Layer:** Patient representation is encoded using diagnosis and procedure information, which are both multi-hot variables. To project a multi-hot variable $\mathbf{d}_i$ or $\mathbf{p}_i$ for a particular visit $i$ into a high-dimensional embedding space, i.e. $\mathbf{e}_d^i$ and $\mathbf{e}_p^i$, we sum the embeddings of the corresponding active elements. For simplicity, we have omitted the patient index (k), the same below. Formally,

$$\mathbf{e}_d^i = \mathbf{d}_i \mathbb{E}_d, \quad \mathbf{e}_p^i = \mathbf{p}_i \mathbb{E}_p, \tag{1}$$

where embedding spaces for diagnosis and procedure are denoted as $\mathbb{E}_d$ and $\mathbb{E}_p$ respectively.

**Long-term Aggregation:** A patient's health status is a dynamic process influenced by various factors such as disease progression and historical treatment records. Relying solely on current symptoms may not provide a comprehensive understanding of the patient's overall condition and disease evolution. Formally, given a patient's clinical history represented as $[\mathbf{d}_1, \mathbf{d}_2, ..., \mathbf{d}_{\mathcal{N}_k}]$, we extract the embeddings $\mathbf{e}_d$ and $\mathbf{e}_p$ and utilize the RNN architecture to capture the visit-level diagnosis and procedure information, respectively. This process can be defined as follows:

$$\mathbf{u}_d^{\mathcal{N}_k} = GRU_d(\mathbf{e}_d^1, \mathbf{e}_d^2, \cdots, \mathbf{e}_d^{\mathcal{N}_k}), \quad \mathbf{u}_p^{\mathcal{N}_k} = GRU_p(\mathbf{e}_p^1, \mathbf{e}_p^2, \cdots, \mathbf{e}_p^{\mathcal{N}_k}) \tag{2}$$

where $\mathbf{u}$ denotes the hidden vector extracted by Gated Recurrent Unit (GRU) and GRU is a two-layer structure. The two vectors will be concatenated to form the patient's final longitudinal representation to query candidate drugs. Formally,

$$\mathbf{u}^{\mathcal{N}_k} = g(\mathbf{u}_d^{\mathcal{N}_k} || \mathbf{u}_p^{\mathcal{N}_k}), \tag{3}$$

where $g(\cdot)$ denotes the transformation matrix and $||$ is the concatenation operation.

## 3.2 MOTIF-AWARE DRUG REPRESENTATION MODULE

In this section, we introduce two modules for enhanced drug characterization: *Drug-centric Representation Module* focuses on uncovering motif interactions within the drug, while *Patient-centric Representation Module* considers the dynamic nature of drug effects, accounting for variations based on the patient's condition over time. Vivid examples of the two modules can be seen in Figure 5 and 6 in the Appendix B respectively.

### 3.2.1 DRUG-CENTRIC REPRESENTATION MODULE

The motif present in a drug holds significant importance in determining its functionality (Zhang et al., 2021), we therefore model drug representation in the motif-level.

**Motif Tree Construction:** To preserve both semantic and structural information to the fullest extent, we initially perform motifs fragmentation using well-defined rules and a strategy involving chemical bond reactions, specifically employing the BRICS method (Ivanov et al., 2023) as an example. This process involves splitting drug molecules based on 16 predefined rules and a set of reactive bonds, while adding dummy nodes to indicate the positions of the motifs. Additionally, we follow MGSSL's post-processing approach to mitigate combinatorial explosion and enhance motif co-occurrence (Zhang et al., 2021). Next, we reassemble the motifs based on the positions of the dummy nodes and record the degree of each node and the frequency of each motif. Using *CNC[C@H](O)C1=CC(O)=C(O)C=C1* as an example, we first construct a molecular object based on its SMILES expression. Then, by applying steps 1) 2) and 3), we fragment the molecule into {'CN':2, 'CC':2, 'CO':3, 'C':1, 'C1=CC=CC=C1':1}, {'CN':2, 'CC':2, 'CO':1, 'C':3, 'C1=CC=CC=C1':3}, resulting in dictionaries that represent the motif recency and degree of motif for each molecule.

**Structure-aware Graph Transformer:** Harnessing the modeling capabilities of graph transformer (Li et al., 2022a; Jiang et al., 2023) to capture intricate relationships between nodes, we employ this approach to capture molecular-level representations. Initially, we count the substructures contained in all drug molecules, recode them into a dictionary $\mathcal{I}$, and assign embedding vectors by $\mathbb{E}_s$. Meanwhile, considering the potential loss of structural information when converting topological data to sequence data, we have innovatively incorporated rich structural embeddings, including degree $\mathbb{E}_{deg}$ and recency $\mathbb{E}_{rec}$, to enhance the completeness of the representation. Formally, for a motif $\mathbf{s}_{i,j}$ contained in drug $i$,

$$\mathbf{e}_s^{i,j} = \mathbf{s}_{i,j} \mathbb{E}_s + W_s(\mathbf{s}_{i,j}^{deg} \mathbb{E}_{deg} + \mathbf{s}_{i,j}^{rec} \mathbb{E}_{rec}), \tag{4}$$

where $\mathbf{e}_s^{i,j}$ denotes the embedding vector of $s_{i,j}$ and $W_s$ is the transformation matrix for structure information. $\mathbf{s}$, $\mathbf{s}^{deg}$, and $\mathbf{s}^{rec}$ represent the one-hot encoding of s corresponding to id, degree and recency. Subsequently, we utilize a graph encoder to capture high-order complex collaborative relationships between drug molecules by incorporating the obtained semantically rich representations. This graph encoder consists of stacked multi-head attention mechanisms, layernorm, and feed-forward layers, ensuring a comprehensive and formal encoding process for the molecular data. Formally, for certain motifs contained in the drug molecular, the update method is as follows,

$$\mathbf{H}_i^{(0)} = [\mathbf{h}_{i,1}^{(0)}, \mathbf{h}_{i,2}^{(0)}, \cdots, \mathbf{h}_{i,\mathcal{N}_s^i}^{(0)}]^\top, \quad \mathbf{H}_i^{(l)} = \text{G-Transformer}\,(\mathbf{H}_i^{(l-1)}), \forall l \in \{1, 2, \cdots, D\}, \quad (5)$$

where $\mathbf{h}_{i,j}^{(0)}$ is initialized with $\mathbf{e}_s^{i,j}$, which refers to the high-level representation of the motif $j$ contained in drug $i$, $\mathcal{N}_s^i$ represents the number of motifs of drug $i$, $D$ represents the number of encoder layers and $\top$ is matrix transpose. Finally, without loss of generality, we use a mean pooling readout function to obtain molecular-level representations, i.e., $\mathbf{h}^i = \text{READOUT}(\mathbf{H}_i^{(D)})$, where $\mathbf{h}^i \in \mathbb{E}_h$ denotes the represenation of drug $i$. Next, we can compute the probabilities for each drug using the patient's longitudinal status. Formally,

$$\mathbf{m}_d^{\mathcal{N}_k} = \text{LN}(\sigma(\mathbf{u}^{\mathcal{N}_k} \mathbb{E}_h^{(\top)})), \quad (6)$$

where $\sigma$ is the sigmoid activation function and LN denotes the layernorm normilization.

### 3.2.2 PATIENT-CENTRIC REPRESENTATION MODULE

Apart from capturing static drug functional representations, we also incorporate patient disease progression to dynamically derive patient-centric functional requirements from two angles: repetition and novel exploration. This motivation is directly inspired by the observations in Appendix D.1.

**Repeat Encoder:** This encoder is utilized to evaluate the probability of re-prescribing a drug with certain motifs, taking into account the patient's disease progression and the functional characteristics of the drug in their historical records. We therefore first sight ingenious ways to characterize the progress of the patient's disease. Formally,

$$\mathbf{q}^u = \mathbf{u}^{\mathcal{N}_k} - \mathbf{u}^{\mathcal{N}_k-1}, \quad (7)$$

where $\mathbf{q}^u$ represents the varying disease. Then we learn a Gaussian distribution (Kingma & Welling, 2013) to stabilize the training process and mitigate noise,

$$\mathbf{q}^u = \text{ReLU}(W_u \mathbf{q}^u), \quad \mu = W_\mu \mathbf{q}^u, \quad \log \sigma = W_\sigma \mathbf{q}^u, \quad \mathbf{z}^u = \mu + \epsilon \odot \sigma, \epsilon \sim \mathcal{N}(0, I), \quad (8)$$

where $\mu$ and $\sigma$ represent the mean and variance, respectively, $\epsilon$ denotes the sampled random value, $W$ denotes the learnable matrix, and $\mathcal{N}(0, I)$ refers to Gaussian distribution. Subsequently, we aggregate all the motifs contained in the previously prescribed drugs for a patient as a functional set $\mathcal{I}_S$ and compute the attention scores between $\mathbf{z}^u$ and each motif. Formally,

$$\mathbf{e}_r^i = \mathbf{v}_r^\top \tanh(W_r \mathbf{e}_s^i + U_r \mathbf{z}^u), \quad P(i \mid r, u, \mathcal{I}_S) = \begin{cases} \frac{\sum_i \exp(\mathbf{e}_r^i)}{\sum_{j=1}^{|\mathcal{I}_S|} \exp(\mathbf{e}_r^j)} & \text{if } i \in \mathcal{I}_S \\ 0 & \text{if } i \in \mathcal{I} - \mathcal{I}_S \end{cases}, \quad (9)$$

where $\mathbf{v}, U$ are learnable vectors and $\mathbf{e}_s \in \mathbb{E}_s$.

**Explore Encoder:** The exploration module is employed to assess the necessity of novel motifs functionalities, denoted as $i \notin \mathcal{I}_S$, thereby recommending new drugs containing such motifs. In this case, due to the progression of the patient's medical condition or changes in symptoms, there is a potential need for additional drugs with new functional properties. Hence, we perform a similar calculation of the patient's medical condition distribution but compute the probabilities associated with all substructures. Formally,

$$\mathbf{e}_e^i = \mathbf{v}_e^\top \tanh(W_e \mathbf{e}_s^i + U_e \mathbf{z}^u), \quad \mathbf{c}_e^{\mathcal{I}_S} = \sum_{i=1}^{|\mathcal{I}_S|} \frac{\exp(\mathbf{e}_e^i)}{\sum_{j=1}^{|\mathcal{I}_S|} \exp(\mathbf{e}_e^j)} \mathbf{e}_e^i, \quad (10)$$

where $\mathbf{c}$ refers to the patient state combined with motifs requirements. Considering that $exp(-\infty) = 0$, we assume that if there exists an item in $\mathcal{I}_S$, the probability of recommending it under the explore mode is zero. Formally,

$$\mathbf{f}_i = \begin{cases} -\infty & \text{if } i \in \mathcal{I}_S \\ W_e^c \mathbf{c}_e^{\mathcal{I}_S} & \text{if } i \in \mathcal{I} - \mathcal{I}_S \end{cases}, \quad P(i \mid e, u, \mathcal{I}_S) = \frac{\exp(\mathbf{f}_i)}{\sum_{j=1}^I \exp(\mathbf{f}_j)}, \quad (11)$$

where $\mathbf{f}$ is equivalent to masking the used motif.

**Discriminate Encoder:** This module calculates the personalized probability of repetition and exploration for the current patient, and the input is the patient's entire historical visit records. We exploit the attention mechanism to synthesize personalized probability distributions. Formally,

$$\mathbf{e}_{re}^i = \mathbf{v}_{re}^\top \tanh(W_r \mathbf{e}_s^i + U_r \mathbf{u}^{\mathcal{N}_k}), \quad \mathbf{c}_d^{\mathcal{I}_S} = \sum_{i=1}^{|\mathcal{I}_S|} \frac{\exp(\mathbf{e}_{re}^i)}{\sum_{j=1}^{|\mathcal{I}_S|} \exp(\mathbf{e}_{re}^j)} \mathbf{e}_{re}^i, \tag{12}$$

$$[P(r \mid u, \mathcal{I}_S), P(e \mid u, \mathcal{I}_S)] = \text{softmax}(W_{re}^c \mathbf{c}_d^{\mathcal{I}_S}),$$

where softmax denotes the activation function. Based on these three modules, we can obtain the repeat and explore probability of each patient-centric motif using the Bayesian formula. Formally,

$$\mathbf{p}_r^u = P(r \mid u, \mathcal{I}_S)P(i \mid r, u, \mathcal{I}_S) \mid i \in |\mathcal{I}|, \quad \mathbf{p}_e^u = P(e \mid u, \mathcal{I}_S)P(i \mid e, u, \mathcal{I}_S) \mid i \in |\mathcal{I}|, \tag{13}$$

where $\mathbf{p}$ denotes the probability vector for all motifs. Furthermore, by incorporating the drug-motif inclusion matrix, we can obtain patient-centric probabilities for all drugs. Formally,

$$\mathbf{m}_{re}^{\mathcal{N}_k} = \mathbf{p}_r^u S_{\mathcal{I}_S} + \mathbf{p}_e^u S_I, \tag{14}$$

where $S_{\mathcal{I}_S}$ and $S_{\mathcal{I}}$ are the drug-motif adjacent matrix of $\mathcal{I}_S$ and $\mathcal{I}$, respectively.

## 3.3 DRUG RECOMMENDATION MODULE

We multiply the Drug-centric drug retrieval probability and the Patient-centric retrieval probability to obtain the recommendation probability for a drug. Formally,

$$\hat{\mathbf{o}}_{\mathcal{N}_k} = \sigma(\mathbf{m}_{re}^{\mathcal{N}_k} \odot \mathbf{m}_d^{\mathcal{N}_k}), \tag{15}$$

where $\odot$ denotes the Hadarmard product (Zhao et al., 2023). In general, the three sub-modules we designed not only integrate the long-term representation of patients, but also obtain recommendation results from three perspectives: functionality, reusability, and exploratory.

## 3.4 MODEL TRAINING

**Accuracy-oriented Loss:** To minimize the gap between *DEPOT* recommendations and the groundtruth, we employ a hybrid loss comprising binary cross-entropy and multi-label margin loss for optimization. Formally,

$$\mathcal{L}_{ACC} = \gamma \mathcal{L}_{bce} + (1 - \gamma) \mathcal{L}_{\text{multi}}, \tag{16}$$

where $\mathcal{L}_{\text{multi}} = \sum_{i,j:m_{\mathcal{N}_k}^i=1, m_{\mathcal{N}_k}^j=0} \frac{\max(0, 1-(\hat{o}_{\mathcal{N}_k}^i - \hat{o}_{\mathcal{N}_k}^j))}{|M|}$, $\mathcal{L}_{bce} = -\sum_{i=1}^{|\mathcal{M}|} [m_{\mathcal{N}_k}^i \log(\hat{o}_{\mathcal{N}_k}^i) + (1 - m_{\mathcal{N}_k}^i) \log(1 - \hat{o}_{\mathcal{N}_k}^i)]$. $m_{\mathcal{N}_k}^i$ is the $i$-th entry of multi-hot target medication set $\mathbf{m}$ (similar define $\hat{o}_{\mathcal{N}_k}^i$ for $\mathbf{o}$), and $\gamma$ is a hyper-parameter, experimentally set to 0.95.

**Safety-oriented Loss:** In addition to addressing accuracy concerns, we also take into account the safety aspect of drug recommendation. Follow (Yang et al., 2021b; Wu et al., 2022b), using the DDI adjacency matrix $\mathcal{A}$, we design the following loss function to ensure the least side effects,

$$\mathcal{L}_{DDI} = \sum_{i=1}^{|\mathcal{M}|} \sum_{j=1}^{|\mathcal{M}|} (\hat{\mathbf{o}}_{\mathcal{N}_k}^{(i)} \cdot \hat{\mathbf{o}}_{\mathcal{N}_k}^{(j)}) \cdot \mathcal{A}_{ij}, \tag{17}$$

where $\hat{\mathbf{o}}_{\mathcal{N}_k}^{(i)} \cdot \hat{\mathbf{o}}_{\mathcal{N}_k}^{(j)}$ computes the combination probability between two drugs. This approach helps mitigate the potential increase in DDI effects within each recommendation set.

**Total Loss:** Given the presence of erroneous prescriptions by doctors, real Electronic Health Record (EHR) data contains a certain level of Drug-Drug Interaction effect (Tan et al., 2022; Shang et al., 2019). Regardless of whether the predictions are correct or incorrect, they have the potential to increase the occurrence of DDIs. To address this issue, we introduce weighting functions that aim to find a solution that improves both prediction accuracy and reduces the DDI effect. Formally,

$$\mathcal{L} = \alpha \mathcal{L}_{ACC} + (1 - \alpha) \mathcal{L}_{\text{DDI}}, \tag{18}$$

Table 1: Performance comparison: MIMIC-III

| Method | Jaccard ↑ | F1 ↑ | PRAUC ↑ | DDI ↓ | Avg.# of Drugs |
|---|---|---|---|---|---|
| K-near | 0.4691±0.0021 | 0.6268±0.0012 | 0.6576±0.0011 | 0.0762±0.0007 | 20.56±0.2836 |
| Logistic Regression | 0.4900±0.0027 | 0.6470±0.0026 | 0.7227±0.0031 | 0.0774±0.0006 | 16.47±0.1091 |
| ECC | 0.4868±0.0016 | 0.6428±0.0030 | 0.7602±0.0024 | 0.0806±0.0011 | 16.01±0.1107 |
| RETAIN | 0.4875±0.0022 | 0.6469±0.0027 | 0.7595±0.0023 | 0.0827±0.0007 | 19.19±0.2303 |
| LEAP | 0.4672±0.0023 | 0.6401±0.0022 | 0.7506±0.0022 | 0.0725±0.0012 | 19.02±0.0508 |
| GAMENet | 0.5092±0.0007 | 0.6654±0.0007 | 0.7675±0.0007 | 0.0806±0.0007 | 25.98±0.2509 |
| MICRON | 0.5170±0.0012 | 0.6722±0.0018 | 0.7628±0.0016 | 0.0719±0.0011 | 18.87±0.1323 |
| SafeDrug | 0.5139±0.0015 | 0.6702±0.0017 | 0.7672±0.0023 | 0.0698±0.0009 | 20.55±0.1632 |
| COGNet | 0.5215±0.0019 | 0.6758±0.0014 | 0.7712±0.0034 | 0.0807±0.0023 | 26.14±0.1061 |
| MoleRec | 0.5306±0.0021 | 0.6843±0.0026 | 0.7737±0.0023 | 0.0726±0.0010 | 20.91±0.1169 |
| DEPOT (**Ours**) | **0.5335±0.0025** | **0.6871±0.0024** | **0.7814±0.0036** | **0.0684±0.0007** | 20.21±0.1887 |

Table 2: Performance comparison: MIMIC-IV

| Method | Jaccard ↑ | F1 ↑ | PRAUC ↑ | DDI ↓ | Avg.# of Drugs |
|---|---|---|---|---|---|
| K-near | 0.3978±0.0023 | 0.5473±0.0022 | 0.3917±0.0023 | 0.0815±0.0013 | 12.78±0.2203 |
| Logistic Regression | 0.4499±0.0033 | 0.5985±0.0027 | 0.7259±0.0024 | 0.0769±0.0008 | 10.36±0.1111 |
| ECC | 0.4350±0.0017 | 0.5809±0.0020 | 0.7193±0.0021 | 0.0681±0.0003 | 8.76±0.2115 |
| RETAIN | 0.4182±0.0020 | 0.5729±0.0016 | 0.6775±0.0012 | 0.0835±0.0004 | 11.29±0.2007 |
| LEAP | 0.4123±0.0022 | 0.5661±0.0019 | 0.6026±0.0025 | 0.0667±0.0005 | 12.13±0.1530 |
| GAMENet | 0.4594±0.0015 | 0.6126±0.0011 | 0.7229±0.0020 | 0.0803±0.0006 | 18.86±0.3302 |
| MICRON | 0.4605±0.0023 | 0.6132±0.0021 | 0.6893±0.0022 | 0.0671±0.0003 | 13.75±0.2172 |
| SafeDrug | 0.4643±0.0035 | 0.6165±0.0020 | 0.7051±0.0011 | 0.0672±0.0004 | 14.06±0.2615 |
| COGNet | 0.4786±0.0010 | 0.6264±0.0011 | 0.6963±0.0018 | 0.0825±0.0002 | 14.71±0.1105 |
| MoleRec | 0.4721±0.0021 | 0.6242±0.0017 | 0.7145±0.0015 | 0.0724±0.0007 | 14.04±0.2203 |
| DEPOT (**Ours**) | **0.4813±0.0019** | **0.6321±0.0015** | **0.7245±0.0020** | **0.0660±0.0005** | 13.19±0.2258 |

where $\alpha$ is are hyper-parameters to achieve a balance. In contrast to prior research that solely examines the DDI rate of the current training batch (Shang et al., 2019), we incorporate the historical DDI to promote training stability and smoothness, thereby mitigating the influence of outliers. Formally,

$$\alpha = \{ \begin{array}{ll} 1 & \kappa \leq \eta \\ \min\{1, \tanh(\tau \frac{\eta}{\kappa - \eta})\} & \kappa > \eta \end{array}, \tag{19}$$

where $\kappa$ is the mean of fixed-length $T$ historical DDI rates. *In this way, our optimization goal is modified to comprehensively extract historical DDI information to ensure that the average value of multiple DDIs is low, thereby ensuring the safety of the model.* The overall optimization process of the algorithm is shown in Algorithm 1 in Appendix C.

## 4 EXPERIMENTS

To demonstrate the state-of-the-art and robustness of our model, we conduct extensive experiments on two extensive real-world Electronic Health Record (EHR) data sets: *MIMIC-III* (Johnson et al., 2016) and *MIMIC-IV* (Johnson et al., 2020). We select some advanced baselines for comparison according to the paradigm to prove the superiority of the proposed *DEPOT*. 1) Rule-based: *K-near* (Shang et al., 2019) 2) Instance-based: *LR* (Luaces Rodríguez et al., 2012), *ECC* (Read et al., 2011) 3) Longitudinal-based: *RETAIN* (Choi et al., 2016), *LEAP* (Zhang et al., 2017), *GAMENet* (Shang et al., 2019), *SafeDrug* (Yang et al., 2021b), *MICRON* (Yang et al., 2021a), *COGNet* (Wu et al., 2022b), *MoleRec* (Yang et al., 2023). The algorithm's performance is assessed using Jaccard, F1-score, PRAUC, and DDI as evaluation metrics. *For crucial data statistics and analysis, baseline mechanism, implementation details and evaluation metrics, please refer to Appendix D. With regard to the necessary hyperparameter tests, please consult Appendix E.*

**Comparison with Baselines:** In Table 1 and Table 2, we present our framework along with the results of all baselines. Overall, *DEPOT* outperforms all other methods on both data sets in terms of accuracy and safety criteria. Compared to MIMIC-III, the performance of all algorithms on the MIMIC-IV data set shows a slight decrease, attributed to the data set's greater diversity and heterogeneity in diagnosis and treatment options, as well as its more complex and challenging drug-disease patterns. In terms of accuracy, the K-Near method performs the worst because it relies on a rule-based approach, neglecting the patient's individual factors and the collaborative filtering effects among drugs. Instance-based methods like LR and ECC show greater improvements in recall and accuracy compared to K-Near, with a nearly 4.5% improvement in jaccard indicators. However,

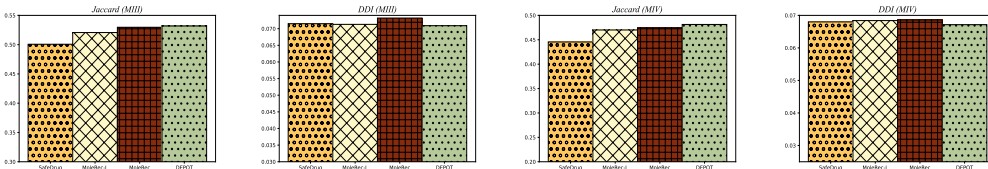

(a) Jaccard of MIMIC-III.    (b) DDI of MIMIC-III.    (c) Jaccard of MIMIC-IV.    (d) DDI of MIMIC-IV.

Figure 3: RECAP Decomposition

due to the lack of consideration for the patient's historical information, they cannot fully capture the patient's state, resulting in significantly lower performance compared to *DEPOT*. Regarding longitudinal-based baslines, RETAIN and LEAP exhibit weaker performance in comparison with SafeDrug and COGNet as they ignores topological information in molecular characterization and high-order similarity. MoleRec, which also models from motifs, is the strongest baseline. However, its effectiveness is still inferior to *DEPOT* because it fails to consider the structural connections between motifs and the relationship between motifs and dynamic disease progression. In terms of safety, the baselines that include DDI regularization, such as GAMENET, MICRON, and COGNET, outperform the baselines that do not consider security criteria, such as K-near, LR, and ECC, on both data sets. Among them, SafeDrug achieves the most competitive DDI rate, which stems from its strict principle of sacrificing recommendation accuracy, that is, only optimizes DDI when the target DDI is exceeded. In contrast, *DEPOT* optimizes the DDI loss in a non-linear form and utilizes past DDI rates to stabilize training, enabling exploration of a broader optimization space. This comprehensive approach allows *DEPOT* to achieve the superior performance in terms of safety, further distinguishing it from other baseline methods.

**Diverse Motif-tree Construction.** To further assess the algorithm's stability, we applied the RE-CAP decomposition (Liu et al., 2017) to reconstruct the motif-tree and conducted comparisons with several competitive algorithms. The results, presented in Figure 3, demonstrate that our algorithm consistently achieves favorable experimental outcomes even under the RECAP decomposition method. It indicates that our approach can effectively handle variations in data decomposition techniques, ensuring consistent performance. This versatility enhances the algorithm's practical utility and suggests its potential for application in different real-world scenarios. Moreover, considering the possibility of future advancements in semantic decomposition methods, our algorithm holds promise for further improvement. By leveraging more sophisticated semantic decomposition techniques, our algorithm has the potential for pluggable enhancements, enabling it to adapt to evolving decomposition methodologies and potentially achieve even better performance.

**DDI Loss:** Figure 4 illustrates the impact of integrating the historical DDI weight determination strategy into SafeDrug (linear), MoleRec (nonlinear), and *DEPOT* (nonlinear) on their DDI loss.

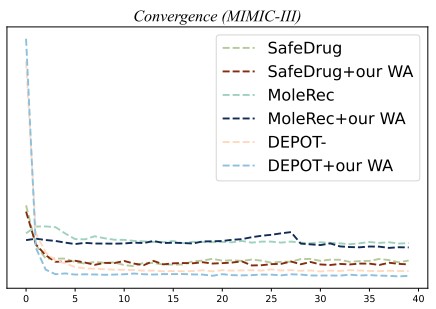

Figure 4: Plug-in DDI Loss. + our WA means using our weighted strategy.

On the one hand, the evidence in the figure demonstrates that the three algorithms exhibit varying degrees of reduction in the lower bound of DDI loss convergence, thus confirming that the incorporation of historical information ensures safer recommendations. On the other hand, the integration of historical DDI enables the three models to achieve a more stable training process, resulting in a smoother loss curve. This stability contributes to more reliable and consistent predictions, ultimately improving the accuracy and robustness of *DEPOT*.

**Ablation Studies:** We further compare *DEPOT* with several ablation variants to demonstrate the effectiveness of different sub-modules. For fairness, we maintain consistent settings across all experiments except for the specific ablation module being tested.

In Table 3, we present the results of different variant methods, showcasing their performance across various metrics. Among the ablation variants, the *DEPOT* stands out as the top performer across all metrics. On the other hand, the *DEPOT*-NE method demonstrates the weakest performance. This is attributed to its limited exploration of the relationships between previously interacted drug motifs, leading to a narrowed recommendation range. The performance of *DEPOT*-NR is less degraded compared to that of *DEPOT*-NE, as it benefits from the collaborative filtering of diseases and drugs

Table 3: Ablation Study: MIMIC-III. (1) **DEPOT-NS** is not aware of motifs collaborations and uses atomic embeddings for graph pooling. (2) **DEPOT-NR** removes the repeat module. (3) **DEPOT-NE** removes the explore module. (4) **DEPOT-NH** has no historical DDI considerations.

| Method | Jaccard ↑ | F1 ↑ | PRAUC ↑ | DDI ↓ | Avg.# of Drugs |
|--------|-----------|------|---------|-------|----------------|
| *DEPOT* | **0.5335±0.0025** | **0.6871±0.0024** | **0.7814±0.0036** | **0.0684±0.0007** | 20.21±0.1887 |
| *-NS* | 0.5309±0.0011 | 0.6841±0.0010 | 0.7799±0.0012 | 0.0688±0.0012 | 21.26±0.2264 |
| *-NR* | 0.5303±0.0015 | 0.6822±0.0012 | 0.7798±0.0013 | 0.0704±0.0006 | 20.26±0.2134 |
| *-NE* | 0.5298±0.0018 | 0.6811±0.0014 | 0.7777±0.0007 | 0.0696±0.0008 | 21.07±0.3001 |
| *-NH* | 0.5313±0.0013 | 0.6856±0.0009 | 0.7797±0.0014 | 0.0696±0.0009 | 20.45±0.3339 |

through a drug-centric module. *DEPOT*-NS performs graph aggregation at the atomic level, limiting its ability to perceive the semantics and synergistic interactions of high-order motifs, thereby resulting in weak performance. Comparing Drug-NH to *DEPOT*, both models exhibit similar accuracy indexes. However, Drug-NH slightly lags behind *DEPOT* in terms of DDI. This discrepancy can be attributed to the impact of DDI outliers in the model.

Overall, each module in our design plays a crucial role in the effectiveness of the *DEPOT* model. The superiority of the *DEPOT* and the relatively weaker performance of the ablation variants highlight the significance of the individual modules and the comprehensive design of our framework.

**Case Study:** To showcase the superiority of *DEPOT* and provide an intuitive understanding, we present the recommendation of *DEPOT*, SafeDrug, and MoleRec for an anonymous patient.

Table 4: Example Recommendation Result

| Method | Recommended Drug Set |
|--------|----------------------|
| **Ground-Truth** Num: 10 | **TP:** ['A06A', 'N02B', 'B01A', 'A01A', 'C02D', 'A04A', 'N02A', 'A12B', 'M03B', 'C08C'] |
| **SafeDrug** Num:12 Recall:0.700 Precision:0.583 | **TP:** ['N02A', 'A12B', 'A04A', 'A01A', 'B01A', 'N02B', 'A06A'] **FN:** ['C02D', 'C08C', 'M03B'] **FP:** ['B05C', 'N05B', 'A12C', 'A02B', 'D04A'] |
| **MoleRec** Num:12 Recall:0.800 Precision:0.667 | **TP:** ['A01A', 'C02D', 'A12B', 'N02B', 'A04A', 'A06A', 'N02A', 'B01A'] **FN:** ['M03B', 'C08C'] **FP:** ['D04A', 'A12C', 'A02B', 'B05C'] |
| **DEPOT** Num:11 Recall:0.800 Precision:0.727 | **TP:** ['A12B', 'N02A', 'C02D', 'A06A', 'B01A', 'N02B', 'A01A', 'A04A'] **FN:** ['M03B', 'C08C'] **FP:** ['A12C', 'A02B', 'B05C'] |

Analyzing the results in Table 4, it becomes evident that *DEPOT* surpasses other methods in both recall and precision. This exceptional performance can be attributed to the fact that *DEPOT* considers the intricate functional collaboration between drug motifs and their relevance to disease processes during the recommendation process. To minimize the occurrence of potent side effects, certain drugs, specifically *M03B* and *C08C*, have been omitted from the recommended results of *DEPOT*. In contrast, SafeDrug overlooks potential drugs due to excessively strict safety guidelines, while MoleRec fails to consider possible drug re-use due to its lack of a dynamic perspective. *We recognize that doctors' levels of experience and expertise can vary, resulting in different prescribed drugs for patients. As a consequence, there is no absolute gold standard for comparison. However, the findings presented strongly suggest that DEPOT holds promise as an advanced drug recommendation system.*

## 5 RELATED WORK

For a thorough comparison with prior studies, we provide a systematic review in Appendix F, covering *drug recommendation* and *molecular characterization*.

## 6 CONCLUSION

In this study, we present *DEPOT*, a drug recommendation framework that leverages drug motifs to provide accurate and safe prescription recommendations to healthcare professionals. Our framework incorporates a structural information-aware module, enhancing the representation of molecular graphs. Additionally, we introduce two modules, repeat and explore, to capture the dynamic relationship between drug motifs and disease progression. Moreover, we integrate historical DDI information in a non-linear fashion to ensure recommendation stability and minimize DDI rates. Through comprehensive comparison and robustness testing on two data sets, we demonstrate that *DEPOT* showcases improved safety and accuracy, thereby assisting healthcare professionals in making informed decisions and paving the way for further advancements in the field of drug recommendation.

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

## A    NOTATIONS USED IN *DEPOT*

We provide a comprehensive list of important mathematical symbols and their corresponding meanings in Table 5 to facilitate better comprehension of the paper.

Table 5: Mathematical Notations

| Notations | Descriptions |
|---|---|
| $\mathcal{X}$ | electronic health record |
| $\mathcal{D}, \mathcal{P}, \mathcal{M}, \mathcal{I}$ | diag, proc, drug, and motif set |
| $\mathbf{d}, \mathbf{p}, \mathbf{m}$ | multi-hot vector of diag, proc, drug, and motif |
| $\mathbf{s}, \mathbf{s}^{deg}, \mathbf{s}^{rec}$ | one-hot vector of motif, degree, and recency |
| $\mathbf{x}$ | concatenation of medical codes |
| $\mathcal{G}$ | ddi graph |
| $\mathcal{V}$ | vertex set same as $\mathcal{M}$ |
| $\mathcal{E}, \mathcal{A}$ | edge set and adjacency matrix of $\mathcal{G}$ |
| $\mathbf{e}_*$ | medical embedding of medical codes $*$ |
| $\mathbf{u}^*$ | patient embedding at $*$ visit |
| $\mathbb{E}_*$ | embedding tables of medical codes $*$ |
| $\mathbf{z}$ | patient disease progression |
| $\mathbf{p}_*$ | probability vector of mode $*$ |
| $S$ | drug-motif adjacent matrix |
| $\hat{\mathbf{o}}$ | recommendation probability vector for drugs |
| $\beta, \alpha$ | hyperparameters for accuracy and safety loss |
| $\eta$ | target ddi value |
| $\kappa$ | mean value of historical ddi rate |

## B  VIVID EXAMPLES

To enhance comprehension, we have created the following illustrations. Figure 5 and 6 elucidate the algorithms presented in section 3.2.1 and 3.2.2, respectively.

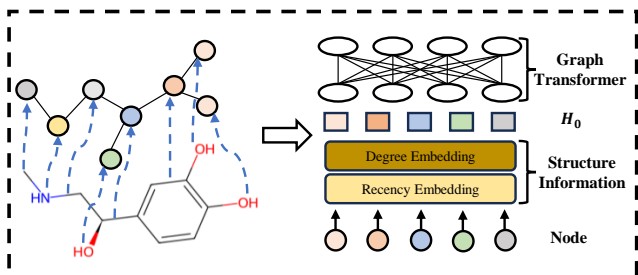

Figure 5: Drug-centric representation module, including motif tree construction and structure-aware graph transformer. $H_0$ denotes the initial embedding.

## C  ALGORITHMS

Algorithm 1 presents a detailed overview of the flow and optimization process employed by the *DEPOT* algorithm.

## D  EXPERIMENT SETUP

In this section, we present *the statistics of the data sets, necessary parameter settings*, and *advanced baselines*.

### D.1  DATA SETS ANALYSIS

We conduct experiments using two extensive real-world Electronic Health Record (EHR) data sets: *MIMIC-III* (Johnson et al., 2016) and *MIMIC-IV* (Johnson et al., 2020). The MIMIC-III data set

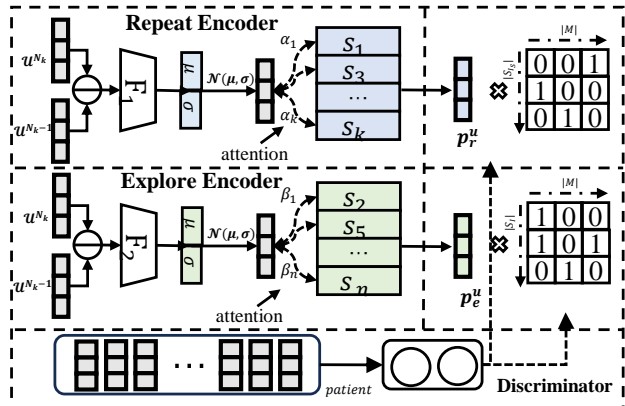

Figure 6: Patient-centric representation module, including repeat encoder, explore encoder and discriminate encoder. The probability output matrix $q$ will be multiplied with the drug-motif adjacency matrix $S$ respectively to obtain the drug recommendation probability.

---

**Algorithm 1** The Algorithm of *DEPOT*

---

**Input:** EHR $\mathcal{X}$, DDI Graph $\mathcal{G}$
**Output:** Parameters $\Theta$;
1: Random initialize model parameters $\Theta$,
2: Data preprocessing $e^d \in \mathbb{E}_d$, $e^p \in \mathbb{E}_p$, $e^s \in \mathbb{E}_s$
3: **while** not converged **do**
4:      Sample a patient $\mathcal{X}^{(k)}$ from $\mathcal{X}$;
5:      **for** $t = 1$ to $\mathcal{N}_k$ **do**
6:          Obtain patient representation $\mathbf{u}$ in Eq. 3
7:          Obtain drug-centric representation $\mathbf{h}$ in Eq. 5;
8:          Calculate drug-centric probability $\mathbf{m}_d$ in Eq. 6;
9:          Obtain disease progression $\mathbf{z}$ in Eq. 8
10:         Obtain repeating probability of the $\mathcal{I}_S$ in Eq. 9
11:         Obtain exploring probability of the $\mathcal{I} - \mathcal{I}_S$ in Eq. 11
12:         Obtain the discrimination probability in Eq. 12
13:         Calculate patient-centric probability $\mathbf{m}_{re}$ in Eq. 14;
14:         Generate the recommendation result $\hat{\mathbf{o}}$ in Eq. 15
15:      **end for**
16:      Evaluate and obtain DDI Rate for the patient
17:      Update the parameters using the loss in Eq. 18
18: **end while**
19: **return** Parameters $\Theta$

---

encompasses data from adult patients in the intensive care unit between 2001 and 2012, while the MIMIC-IV data set comprises comprehensive electronic medical records and clinical information of hospitalized patients between 2008 and 2019. It is important to note that these data sets are fully anonymized and subjected to privacy protection measures before our access. Table 6 provides detailed statistics regarding the data sets, offering insights into patient numbers, diagnosis codes, and medication usage.

In order to test our hypotheses and intuitions, we perform preliminary data analysis on the data sets. As depicted in Figure 7(a), we notice that drug recommendation scenarios require the recommendation of a collection of drugs of various sizes, rather than a single item, which represents a significant departure from traditional recommendation mechanisms. Figure 7(b) displays the proportion distribution of historical drug reuse, while Figure 7(c) presents the distribution of historical motif reuse. Both distributions demonstrate a right-skewed pattern, providing compelling evidence for the importance of leveraging repetition and exploration perspectives. *In a nutshell, these data analysis results serve as an empirical foundation for our research, ensuring the reliability and feasibility of our experiments.*

Table 6: Data Statistics

| Items | MIMIC-III | MIMIC-IV |
|---|---|---|
| # of patients / # of visits | 6,350 / 15,032 | 61,267 / 163,875 |
| diag. / prod. / drug. set size | 1,958 / 1,430 / 130 | 2,000 / 11,055 / 130 |
| avg. / max # of visits | 2.3672 / 29 | 2.6748 / 70 |
| avg. / max # of diag per visit | 10.5089 / 128 | 8.2345 / 270 |
| avg. / max # of prod per visit | 3.8437 / 50 | 2.3579 / 95 |
| avg. / max # of drug per visit | 11.4296 / 65 | 6.4889 / 72 |
| total # of DDI pairs | 674 | 800 |
| total # of motifs | 225 | 225 |

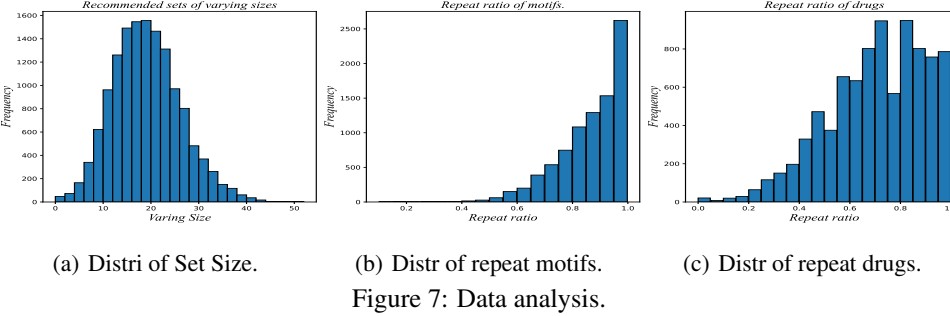

(a) Distri of Set Size.  (b) Distr of repeat motifs.  (c) Distr of repeat drugs.

Figure 7: Data analysis.

## D.2 BASELINES

In line with the genre in related work, we have chosen several advanced baselines for comparison.

- **K-Near (Shang et al., 2019):** Exactly the same medications recommended for the current visit as prescribed for the last visit.

- **LR (Luaces Rodríguez et al., 2012):** LR is a traditional recommendation technique with L1 regularization, we train a separate classifier for each class for multi-label classification.

- **ECC (Read et al., 2011):** ECC employs boosting-based ensemble learning for multi-label classification, where the output of the previous base classifier serves as a feature input for the subsequent classifier.

- **RETAIN (Choi et al., 2016):** RETAIN integrates attention mechanism and RNN with the GRU architecture to represent historical electronic medical records.

- **LEAP (Zhang et al., 2017):** LEAP formulates drug recommendation as a sequential decision problem, utilizing a recurrent decoder and an attention mechanism to capture label dependencies.

- **GAMENet (Shang et al., 2019):** GAMENet incorporates DrugGraph and EHR Graph in the longitudinal modeling process to ensure the effectiveness and safety of recommendations.

- **MICRON (Yang et al., 2021a):** MICRON leverages residual-based inference to link drug changes and disease changes, enabling incremental learning of new patient characteristics.

- **SafeDrug (Yang et al., 2021b):** SafeDrug utilizes the atomic architecture of drug molecules for characterization and designs mask matrices to explore the relationship between molecules and diseases. We use the linear adjusting version.

- **COGNet (Wu et al., 2022b):** COGNet retrieves his or her historical diagnosis and drug recommendations, and mines their relationship with the current diagnosis, which is embedded in Transformer as a plug-in for generative inference.

- **MoleRec (Yang et al., 2023):** MoleRec constructs substructure embeddings according to the inclusion relationship of molecules and substructures, and designs an attention mechanism to capture their interaction relationship with diseases.

Please Note that these algorithms are categorized as follows, 1) Rule-based: *K-near* 2) Instance-based: *LR, ECC* 3) Longitudinal-based: *RETAIN, LEAP, GAMENet, SafeDrug,MICRON, COGNet, MoleRec*. To be fair, we tune the hyper-parameters of each model to achieve the best results.

### D.3 IMPLEMENTATION DETAILS AND EVALUATION PROTOCOLS

Our code has been open-sourced and can be found on GitHub [2]. Regarding the data set, we divided it into three subsets: training set, validation set, and test set as previous (Shang et al., 2019; Yang et al., 2021b; 2023). The training set accounts for 2/3 of the data set, while the validation and test sets each comprise 1/6 of the data set. Our model was implemented using PyTorch and trained on an Ubuntu system with a GeForce RTX 3090ti GPU. The initial embeddings were initialized using Kaiming initialization (He et al., 2015) and had a dimension of 64, unless specified otherwise for concatenation operations. To optimize the model during training, we employed the Adam optimizer with a learning rate of 5e-4. We conducted hyperparameter tuning on relevant parameters, including parameter $dim$ within $[32, 48, 64, 96]$, parameter $l$ within $[1, 2, 3, 4]$, parameter $T$ within $[2, 3, 4, 5]$, and parameter $\tau$ within $[0.5, 0.6, 0.8, 1.0]$ .

In evaluating the performance of our recommendation model, we utilize several metrics, including Precision-Recall AUC (PR-AUC), Jaccard Similarity Score, and average F1 score (F1). These metrics provide insights into the model's accuracy and performance. Additionally, we consider the drug-drug interaction rate (DDI Rate) as a crucial safety evaluation metric, where a lower DDI Rate indicates a safer recommendation in terms of drug-drug interactions.

The following provides detailed definitions of each metric used to evaluate the performance of all algorithms. *The Jaccard coefficient is a mathematical metric used to assess the similarity between two sets.* Formally,

$$
\begin{aligned}
\text{Jaccard} &= \frac{1}{\mathcal{N}_k} \sum_{t=1}^{\mathcal{N}_k} \text{Jaccard}^{(t)} \\
&= \frac{1}{\mathcal{N}_k} \sum_{t=1}^{\mathcal{N}_k} \frac{|\{i : \hat{\mathbf{o}}_i^{(t)} = 1\} \cap \{i : \mathbf{o}_i^{(t)} = 1\}|}{|\{\hat{\mathbf{o}}_i^{(t)} = 1\} \cup \{i : \mathbf{o}_i^{(t)} = 1\}|},
\end{aligned}
\tag{20}
$$

where $\mathcal{N}_k$ refers to the total number of visits of the patient, $t$ represents the $t$-th visit, $\mathbf{o}_i$ and $\hat{\mathbf{o}}_i$ refer to the $i$-th entry of the groudtruth and prediction results, respectively. Please note that the final result is obtained by calculating the mean of the Jaccard coefficients across all visits for each patient.

*The F1-score is a single metric that combines precision and recall to provide a balanced measure of a model's performance.* Formally,

$$
\begin{aligned}
\text{Precision}^{(t)} &= \frac{|\{i : \hat{\mathbf{o}}_i^{(t)} = 1\} \cap \{i : \mathbf{o}_i^{(t)} = 1\}|}{|\{i : \hat{\mathbf{o}}_i^{(t)} = 1\}|}, \\
\text{Recall}^{(t)} &= \frac{|\{i : \hat{\mathbf{o}}_i^{(t)} = 1\} \cap \{i : \mathbf{o}_i^{(t)} = 1\}|}{|\{i : \mathbf{o}_i^{(t)} = 1\}|},
\end{aligned}
\tag{21}
$$

$$
\text{F1-score} = \frac{2}{\mathcal{N}_k} \sum_{t=1}^{\mathcal{N}_k} \frac{\text{Precision}^{(t)} \, \text{Recall}^{(t)}}{\text{Precision}^{(t)} + \text{Recall}^{(t)}}.
\tag{22}
$$

*PRAUC assesses the overall effectiveness of the model across various thresholds of the decision boundary in terms of precision and recall.* Formally,

$$
\text{PR-AUC} = \frac{1}{\mathcal{N}_k} \sum_{t=1}^{\mathcal{N}_k} \sum_{k=1}^{|\mathcal{M}|} \text{Precision}(k)^{(t)} \Delta \text{Recall}(k)^{(t)}
\tag{23}
$$

where $\Delta \text{Recall}(k)^{(t)} = \text{Recall}(k)^{(t)} - \text{Recall}(k-1)^{(t)}$.

*DDI is used to measure the safety of recommendations, that is, the proportion of side effects contained in recommendation results.* Formally,

$$
\text{DDI} = \frac{\sum_{i=1}^{\mathcal{N}_k} \sum_{a,b \in \{j : \hat{\mathbf{o}}_j^{(t)} = 1\}} \mathbb{I}\{\mathcal{A}_{a,b} = 1\}}{\sum_{i=1}^{\mathcal{N}_k} \sum_{a,b \in \{j : \hat{\mathbf{o}}_j^{(t)} = 1\}} 1},
\tag{24}
$$

where $\mathbb{I}$ is an indicator function that equals 1 when $\mathcal{A}_{a,b}$ is 1, and 0 otherwise.

---

[2]https://anonymous.4open.science/r/DrugRec

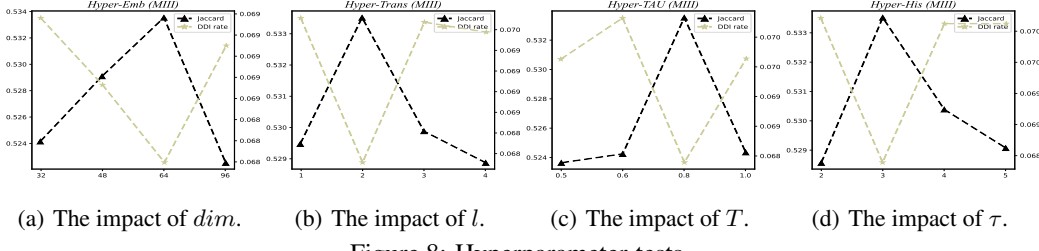

(a) The impact of $dim$.   (b) The impact of $l$.   (c) The impact of $T$.   (d) The impact of $\tau$.

Figure 8: Hyperparameter-tests.

## E   HYPER-TESTING

In this subsection, we present the tuning of several key hyper-parameters in our framework.

**Embedding size *dim*.** To examine the impact of embedding dimension on the performance of our proposed model, we conduct scaling experiments by varying the embedding size. The results are presented in Figure 8(a). The findings demonstrate that *DEPOT* initially experiences improvement as the embedding dimension increases. This improvement continues until reaching a peak when the dimension is set to 64. However, when the dimension exceeds 64, there is a significant drop in performance. This decline can be attributed to the introduction of noise and the potential for overfitting in the model when the dimensionality becomes too large. Emperimentally, we set $dim = 64$.

**Number of Transformer Layers $l$.** The number of Transformer layers in our model plays a crucial role in capturing higher-order collaboration between motifs. Figure 8(b) illustrates the results of our experiments, indicating that the best performance is achieved when $l = 2$ (the optimal number of layers). When the number of layers is too small, the model may struggle to effectively capture the structural relationships between motifs. This limitation hampers the model's ability to understand complex interactions and may result in suboptimal performance. Conversely, when the number of layers is excessively large, the performance improvement may reach saturation, making further increases in the number of layers unnecessary and inefficient.

**History Length $T$.** When the historical length is longer, the calculated historical indicator rate is more stable and accurate. The calculation of the average value is based on a larger number of data points, providing a better reflection of the trends in historical drug interactions. However, there may be some potential issues with excessively long historical lengths, such as increased computational complexity and data storage requirements. Moreover, longer historical lengths may include outdated or less representative data, which can have a negative impact on the accuracy of the results. As shown in Figure 8(c), we observe better performance when $T = 3$.

**Temperature $\tau$.** $\tau$ controls the nonlinearity of the tanh function, which affects the level of constraint on DDI effect. A smaller value of $\tau$ causes the curve to rise more rapidly when it surpasses the target DDI, thereby assigning higher weight to DDI optimization. Conversely, a larger value of $\tau$ results in a more gradual curve and less emphasis on DDI optimization. As illustrated in Fig. 8(d), our model achieves superior performance on both accuracy and safety objectives when $\tau = 0.08$.

## F   RELATED WORK

To provide readers with a comprehensive understanding of the advancements in the field and establish the connections and distinctions between our work and other studies, this section presents a systematic review of the most pertinent literature in the domains of *drug recommendation* and *molecular characterization*.

### F.1   DRUG RECOMMENDATION

Drug recommendation, as a challenging sub-problem in the field of smart healthcare, serves as an essential tool to assist doctors in making prescriptions and alleviate their workload. In contrast to traditional recommendation systems (Ren et al., 2019; Zhao et al., 2023), drug recommendation systems face a dual challenge of *recommending target sets of varying sizes* and *adhering to stringent safety criteria* (Yang et al., 2021b; He et al., 2018; Zheng et al., 2023).

In the early stages of research in the field of drug recommendation, manual rule-based methods were commonly employed. For instance, methods like K-freq (Wang et al., 2019) and K-Near (Shang et al., 2019) relied on selecting drugs based on their symptom frequency or the drug set used in previous visits. However, these approaches heavily relied on the expertise of doctors and lacked flexibility in adapting to individual patient needs. Inspired by recommender systems used in e-commerce (Wu et al., 2022a), instance-based methods were introduced to enhance the efficiency and accuracy of drug recommendation. These methods leveraged the co-occurrence relationship between drugs and patients, akin to collaborative filtering algorithms. For example, MedRec (Zhang et al., 2023) constructed a knowledge graph linking symptoms and drugs and utilized link prediction techniques. ECC (Read et al., 2011) formulated the recommendation task as a multi-label classification problem and employed chained ensemble classifiers to improve accuracy. Subsequently, researchers explored drug recommendation from a longitudinal perspective, taking into account patient history and temporal dependencies among drugs. GameNet (Shang et al., 2019) employed recurrent neural networks (RNNs) to model the patient's historical states and combined electronic health records (EHRs) with drug-drug interaction (DDI) graphs to explore co-occurrence relationships among drugs. SafeDrug (Yang et al., 2021b) integrated drug molecular graphs into the representation module to enhance the drug-side representation. COGNet (Wu et al., 2022b) retrieved a patient's historical diagnoses and drug recommendations, and mined their relationship with the current diagnosis to generate personalized recommendations. Safety considerations in drug recommendations also gained attention to promote the preference for safer drug combinations. For instance, 4SDrug (Tan et al., 2022) incorporated small and safe principals into the training process of the recommendation model, utilizing a mixed penalty loss to ensure that the generated drug combinations carried lower risk.

*Despite decent performance, these methods often overlook crucial motifs information. One relevant work in this regard is MoleRec (Yang et al., 2023), which primarily focuses on the inclusion relationship between drugs and substructures. While this approach captures some aspects of motifs information, it fails to fully explore the structural intricacies between motifs and their dynamic connections with changes in patient diseases. Our proposed framework, DEPOT, takes this into account by introducing motif trees, allowing us to capture essential motifs information for more robust molecular characterization. In addition, we incorporate the relationship between longitudinal disease progression and drug structure, a factor that is not accounted for in MoleRec.*

### F.2 MOLECULAR CHARACTERIZATION

Effective characterization of drugs is crucial for enhancing the performance of drug recommendations. Currently, there are two main streams of molecular modeling methods: *sequence-based models* and *graph-based representation models* (Atz et al., 2021; Sun et al., 2022b; Pattanaik & Coley, 2020).

Sequence-based models drew inspiration from successful approaches in natural language processing. These models could capture the statistical features and semantic information of molecular SMILE sequences, enabling the understanding of molecular structures and properties. For instance, recurrent neural networks (RNNs) or transformers could be employed to model SMILES sequences and make predictions on molecular properties (Zheng et al., 2019; Huang et al., 2021). On the other hand, graph-based models represented molecules as graphs, where atoms or groups are nodes, and chemical bonds are edges (Yi et al., 2022; Fang et al., 2022). This approach incorporated non-Euclidean geometry data mining and took into account the connections, topological relationships, and chemical bond information between atoms. Graph convolutional neural networks (GCNs) or graph attention networks (GATs) were commonly utilized to model molecular graphs and predict molecular properties (Li et al., 2022b; Xu et al., 2021). In recent years, efforts have been made to unify sequence-based and graph-based modeling methods by representing molecular graphs as atomic sequences and utilizing positional encoding to incorporate coordinate information (Li et al., 2022a; Wu et al., 2023a).

*In our study, we draw inspiration from molecular characterization and employ various chemical decomposition methods to divide drug molecules into meaningful motifs. We consider the structural associations between these motifs at a higher level and integrate them into the graph transformer model. This approach provides a richer and more comprehensive molecular characterization, thereby improving the performance of drug recommendations.*

