# OpenReview forum: "Enhancing Precision Drug Recommendations via Fine-grained Exploration of Motif Relationships"
_ICLR.cc/2024/Conference — ICLR 2024 Conference Withdrawn Submission_

### Official Review · Reviewer_Wx6A · 2023-10-30

**Soundness:** 3 good
**Presentation:** 2 fair
**Contribution:** 2 fair
**Rating:** 5
**Confidence:** 3

**Summary:**

This paper claims the existing approaches in drug recommendation neglect to effectively leverage drug motifs information. To leverage the motifs information, this paper proposes to construct drug motif trees and applies graph transformer to learn drug representation. The drug representations are then used to identify the relevancy to patient's current condition as well as to predict the new possible drugs. Finally, drug-drug interaction information is used to reduce the risk that could potentially caused by harmful drug-drug interactions. This paper reports that it out performs several non-trivial baselines on two datasets on drug recommendation task. Module level ablation study as well as studies on motif-tree construction and DDI loss are provided.

__Initial recommendation__: Weak reject based on the weak points.

**Strengths:**

* The paper's motivation, positioning as well as the proposed four-module approach are sound to me.
* The empirical comparison supports the effectiveness claim of the proposed approach.
* The ablation studies show module designs indeed contributes to the final performance.

**Weaknesses:**

* The description in the method section is confusing.
	* The degree and recency variable are not defined, it is unknown what are the domains they belong to.
	* The notation used in Eq(6) overloads the notation of multihot medication records in Section 2 without explanation.
	* The functional set $\mathcal{I}_S$ is not formally defined.
	* The shape of $e_r^i$ is not defined in eq.9, but inferred by the probability definition of $P(i|r, u, \mathcal{I}_S)$, $e_r^i$ seems to take scalar-value. This seems not align with the shape of similarly defined (notation-wise) $e^i_e$ which inferred from eq 10 seems to be a vector.
	* The text above eq.12 mentioned the input of the module is patient's entire historical visit records, however only record $k$ is appeared in the eq.12.
	* Are the two term drug-motif inclusion matrix and drug-motive adjacency matrix referring to the same entities in Eq 14?
	* In section 3.4, first line,  "the ground truth" is not previously defined.
	* Line below 17,  "the combination probability" is not previously defined.
	* The share of the input and output variable are in general missing from the description, which makes the soundness checking of the technical part difficult.
* Multiple technical choices made in the paper are not well motivated.
	* For example, as diagnosis and procedure are multihot variables, the paper uses sum pooling to obtain the corresponding embeddings. A potential side effect of this approach is the resulted embeddings' norm are biased and correlated to the cardinality of the multihot variables. Whether such effect is the favorable to the paper is unknown.
	* Also, the varying disease representation in eq 7 only takes one step of in the user's history thus implicitly assume the disease representation is Markovian, which is not motivated.
	* The two loss term used in the paper in Eq 16 are presented as is without motivation.
* Important submodule in the method is not motivated and described in the paper's introduction section. In section 3.2, the a submodule named "discriminate encoder" is described. According to the math, this module learns to weigh the importance of repetition and exploration preference of a patient. As this module was neither motivated in the introduction, nor examined in the ablation study, its necessity is unclear.

__Additional feedbacks__
* Eq 7, "-1" is a subscript.
* Figure 2, the dashed line of embedding $E_h$ connecting module (b) could be made more logical by swapping the embedding $E_h$ in module (b) to the top of the figure.

**Questions:**

* If the comments about the presentation of the method section are sensible, could you provided a revision plan addressing the issues.
* Could you provide the motivation for the technical choices mentioned in the weakness points?
* Could you adjust the narrative of the paper to motivate the discriminate encoder module?

---

### Official Review · Reviewer_eiqZ · 2023-11-01

**Soundness:** 2 fair
**Presentation:** 2 fair
**Contribution:** 2 fair
**Rating:** 5
**Confidence:** 3

**Summary:**

In this paper, the authors present a novel approach for enhancing the precision of drug recommendation by conducting a thorough exploration of motif relationships, which they call DEPOT (Drug rEcommendation via fine-grained exploration of mOtif relaTionships). DEPOT is a drug recommendation framework that utilizes drug motifs to deliver accurate and safe prescription recommendations. This work includes a structural information-aware module that enhances the representation of molecular graphs. Additionally, the authors have incorporated two modules, known as the repeat and explore encoders, in an attempt to capture the dynamic relationship between drug motifs and disease progression. These three modules are designed to integrate long-term patient representations, offering recommendation results from three distinct perspectives: functionality, reusability, and exploratory. Moreover, historical Drug-Drug Interaction (DDI) information is integrated in a non-linear way to ensure recommendation stability and minimize DDI rates. The authors conducted experiments on two datasets, and the results demonstrate that DEPOT exhibits a modest performance improvement over the baseline methods.

**Strengths:**

This paper presents a novel combination of existing ideas, enhancing the representation learning aspect of prior research by employing motifs as higher-level structures. The novel representation learning technique aims to capture critical characteristics, such as disease progression and drug motifs, and their intricate interplay. The experiments in the paper are comprehensive, following a well-defined methodology based on peer-reviewed papers. Additionally, the authors share their code, ensuring the reproducibility of their proposed method.

**Weaknesses:**

The paper needs to be proofread for English language and checked for typos. Although the paper overall is well written, I struggled following the proposed method section. This section requires significant improvement for clarity.  Specifically, I felt that sometimes there was an unnecessary addition of technical detail and an overall lack of intuition provided. The notation also needs refinement; there are instances of variable reuse and repurposing for different meanings, along with some undefined variables. Specific examples illustrating these concerns are provided in the Questions section of this review.

My biggest concern is that I believe that the experimental results are not strong enough to substantiate the claims made throughout the paper. DEPOT showcases only marginal improvements over the state-of-the-art (SOTA), with gains of no more than 0.01 across all metrics, and often even less. A similar trend is observed in the ablation studies, where the incremental improvements do not seem commensurate with the inclusion of each component. Notably, in the ablated models, the performance remains comparable to, or in some cases even better than, the SOTA.

There are claims like this: “DEPOT-NS performs graph aggregation at the atomic level, limiting its ability to perceive the semantics and synergistic interactions of high-order motifs, thereby resulting in weak performance.” However, the performance of DEPOT-NS was, at its worst in one of the metrics, 0.003 lower than DEPOT. This marginal difference in performance raises questions about whether it is appropriate to label a model without motif collaboration as having 'weak performance' based on such a small variation.

Given that the venue is intended for a broader audience, which may include readers unfamiliar with the specific subfield presented in this paper, it is crucial to provide a more comprehensive context. A prime example of this issue can be found in the 'Diverse motif tree construction' section, where the authors introduce the 'RECAP decomposition' method. While the paper includes relevant citations, it is essential to supplement this with explanations or intuitive descriptions of the RECAP decomposition method for the benefit of readers unfamiliar with it. Additionally, there is a lack of clarity regarding the significance of the results presented in Figure 3 supporting the claims made in the respective section. Figure 3 lacks a proper caption and is not explained in the text at all, making it challenging for readers to discern its relevance and implications.

Minor comments:
- A refinement of the captions would benefit the clarity of the paper.
- For Figure 2, $m_{re}^{N_k}$ and $m_{d}^{N_k}$ have different lengths but should be the same considering the Hadamard product. Also, the font style should follow the math style.
- In Section 3.2.1., the acronym MGSSL needs to be defined.
- In Figure 7(a), the label varing needs to be changed to varying.
- On page 5, normilization to normalization.
- There are some references that point to pre-prints instead of the venues where they were published:
    - Diederik P Kingma and Max Welling. Auto-encoding variational bayes. arXiv preprint  arXiv:1312.6114, 2013.
	  - It was published in ICLR 2014
    - Chaoqi Yang, Cao Xiao, Lucas Glass, and Jimeng Sun. arXiv preprint arXiv:2105.01876, 2021a.  Change matters: Medication change prediction with recurrent residual networks. arXiv preprint arXiv:2105.01876, 2021a.
	  - It was published in the Proceedings of the Thirtieth International Joint Conference on Artificial Intelligence (IJCAI-21)
    - Chaoqi Yang, Cao Xiao, Fenglong Ma, Lucas Glass, and Jimeng Sun. Safedrug: Dual molecular graph encoders for recommending effective and safe drug combinations.  arXiv preprint arXiv:2105.02711, 2021b.
	  - Proceedings of the Thirtieth International Joint Conference on Artificial Intelligence (IJCAI-21)

**Questions:**

Regarding notation:
- In Equation 3, $g( \cdot )$ is used to denote a transformation matrix, however, in Equation 4, $W$ is used to denote a different transformation matrix. It is unclear to me why the different conventions to denote this. Perhaps it would be better to maintain one convention of notation throughout the paper.
- Right above Equation 6, $h^i$ is defined, but the $i$ for drugs was used as a subscript before. Is this correct?
- In Equations 7 and 8, $q^u$ is reused. This part is not entirely clear to me. It is being used in its own definition, which makes me think there’s a sequential order to these equations. Can the authors provide some more details about this?
- In Equation 19, what is $\eta$? I don’t see it defined.

In Figure 2, what is Func Cross? I don’t see it in the caption or referenced in the text. Can the authors elaborate on this?

I have a question regarding Equation 15 in the paper, where the authors employ the Hadamard product to combine the patient-centric and drug-centric retrieval probabilities and subsequently apply the sigmoid function. Given that the product of two probabilities always yields a value between 0 and 1, it appears that the lowest possible input to the sigmoid function can be 0. This implies that the range of Equation 15 and the output $\hat{o}_{N_k}$ will be $[0.5, 1)$, rather than the desired $(0, 1)$. Could you please clarify if you have taken this into consideration? If not, it's important to note that having the lowest probability set to 0.5 could potentially have a significant impact on the model's training process.

For the DDI loss section, what is WA in Figure 4? I don’t believe this is mentioned and it should be clarified.

I have some reservations about the Case Study presented in the paper, particularly regarding the potential for 'cherry-picking' this specific example. It seems that the intended purpose of this section is to demonstrate DEPOT's superiority in terms of Recall and Precision (however, the assertion about Recall is not entirely accurate since DEPOT has the same Recall as MoleRec). To provide more robust evidence in this regard, would it be beneficial to consider a more general approach? For instance, calculating the average recall and precision across all patients in the test set might offer a more comprehensive and representative assessment of DEPOT's recommendation performance relative to other methods.

---

### Official Review · Reviewer_XZo3 · 2023-11-03

**Soundness:** 1 poor
**Presentation:** 2 fair
**Contribution:** 2 fair
**Rating:** 3
**Confidence:** 4

**Summary:**

In this paper, the authors present DEPOT, a deep learning approach for medication recommendation. DEPOT incorporates motifs, which are substructures of drugs, as higher-level structures to improve the recommendation process, an aspect that has been relatively unexplored in previous methods. By integrating static drug functional representations and patient representations through a repetition and exploration module, DEPOT achieves superior performance. The effectiveness of DEPOT is demonstrated through extensive experiments conducted on two widely recognized benchmark datasets, where it outperforms existing methods in medication recommendation.

**Strengths:**

1. The task of medication recommendation holds significant importance in the healthcare domain, making it imperative to explore the potential of motif information in improving its efficacy.

2. The conducted experiments encompass a comprehensive evaluation using two widely recognized electronic health record (EHR) datasets, alongside multiple relevant benchmark techniques, thereby ensuring a rigorous analysis.

**Weaknesses:**

1. The paper's presentation and clarity could benefit from further refinement to enhance its comprehensibility and readability.

2. The proposed method heavily relies on historical medical conditions, which poses limitations in handling patients without prior background information.

3. The absence of a detailed case study showcasing the benefits of incorporating motif information in medication recommendation weakens the persuasiveness and effectiveness of the proposed approach.

**Questions:**

1.	Motif, which is a key concept in this paper, should be formally formulated in sec. 2.

2.	Patients without historical information are prevalent in EHR datasets. Therefore, it’s important to discuss how can the proposed method adapt to these “new” patients.

3.	There is some unclear expression:
a)	In 3.2.1. Motif Tree Construction
i.	“Then, by applying steps 1)2)and3)”, what are the steps?
ii.	“we fragment the molecule into {…}, {…}”, what is the meaning of the two different fragment result? What is the structure of the motif “Tree”?
b)	In 3.2.1. Structure-aware Graph Transformer
i.	What’s the definition of ‘degree’ and ‘recency’ of motif, and why they are important for motif representation?
ii.	Why a transformation matrix W_s is needed in formula (4)?
c)	In 3.2.2
i.	The model design of Repeat and Explore Encoder is similar. Why does the explore encoder also take the “varying disease” q^u as input, rather than current health condition?
ii.	In equation (8): It’ s confusing to take q^u as input and output of the same equation. What’s the meaning of z^u?

3.	How and why can the introduced Gaussian distribution stabilize the training process and mitigate noise?
Overall, this subsection lacks high level insight about the rationality of the model design and the notation is too complex to understand.
iii.	In experiment, the meaning and necessity of Avg. # of Drugs should be clear.
iv.	The font size in figures is too small to recognize, e.g., Fig 3, 7
v.	In page 8, DDI loss: it’s meaningless to show ddi rate without showing accuracy at the same time. The claimed “smoother loss curve” should be supported by a quantitative index.

4.	It’s recommended to add case study to show how the motif benefits the medication recommendation, e.g., by recommending a motif that is important to treat a disease, which cannot be captured by previous method.

---

### Official Review · Reviewer_pMng · 2023-11-04

**Soundness:** 2 fair
**Presentation:** 2 fair
**Contribution:** 2 fair
**Rating:** 3
**Confidence:** 3

**Summary:**

This paper investigates a drug recommendation problem, i.e., how to recommend drug in order to minimize the drug-drug interaction (DDI) rate. In this paper, the authors consider the drug recommendation triplet to formalize the problem, and first learns the patient's representations from the record + state information, and then combines the learned patient's representations with some information from the motif side. Meanwhile, this paper learns the relationship between a patient and a motif using the attention mechanism, and tries to generalize it to the unoccurring motif. Experiments on two real-world Electronic Health Record (EHR) datasets demonstrate the effectiveness of the proposed method.

**Strengths:**

S1: This paper provides a complete process and framework for the drug recommendation step by step.

S2: This paper uses repeat encoder and explore encoder to capture the correlation between patient and motif, which is a novel perspective.

S3: This paper has a comprehensive related work review and baselines.

**Weaknesses:**

W1: The presentation of the paper is poor and the notation is confusing. For example, since the paper contains a large number of notations , the authors could consider providing some information about the dimensionality of the vectors. In addition, in the experiment, the expression and font size in Figure 3 make it difficult to read, and the title of vertical axis in Figure 4 is missing.

W2: This paper contains many steps, the plausibility of which should be verified with much more experiments. For example, the authors claim that they have innovatively incorporated rich structural embeddings, including degree $E_{deg}$ and recency $E_{rec}$. The validity of this step needs to be verified by ablation studies. In addition, it is possible to add another layer of feed-forward neural network instead of directly performing the multiplication in Eq. (6) and Eq. (15).

W3: Why is only consider the last state $N_{k}$ in stead of all the states in some equations like Eq. (7) and Eq. (15)? Is it possible to combine attention or weight decay technic to weight all states to obtain more information?

W4: In explore encoder part, why are the numerator and denominator of Eq 10 is exp($e_{e}$) instead of exp($e_{r}$)? A representation learned using the observed motif should be more helpful.

W5: what is the usage of the discriminate encoder? In addition, what is the meaning of P(r | u, $I_{S}$), P(e | u, $I_{S}$) and Eq. (13)?

**Questions:**

Please refer to the weakness part for the questions.